# A Review of Resistance to Polymyxins and Evolving Mobile Colistin Resistance Gene (*mcr*) among Pathogens of Clinical Significance

**DOI:** 10.3390/antibiotics12111597

**Published:** 2023-11-06

**Authors:** Shakeel Shahzad, Mark D. P. Willcox, Binod Rayamajhee

**Affiliations:** School of Optometry and Vision Science, University of New South Wales, Sydney, NSW 2052, Australia; b.rayamajhee@unsw.edu.au

**Keywords:** polymyxin resistance, molecular evolution, resistance mechanisms, *mcr*, phylogeny

## Abstract

The global rise in antibiotic resistance in bacteria poses a major challenge in treating infectious diseases. Polymyxins (e.g., polymyxin B and colistin) are last-resort antibiotics against resistant Gram-negative bacteria, but the effectiveness of polymyxins is decreasing due to widespread resistance among clinical isolates. The aim of this literature review was to decipher the evolving mechanisms of resistance to polymyxins among pathogens of clinical significance. We deciphered the molecular determinants of polymyxin resistance, including distinct intrinsic molecular pathways of resistance as well as evolutionary characteristics of mobile colistin resistance. Among clinical isolates, *Acinetobacter* stains represent a diversified evolution of resistance, with distinct molecular mechanisms of intrinsic resistance including *nax*D, *lpx*ACD, and *stk*R gene deletion. On the other hand, *Escherichia coli*, *Klebsiella pneumoniae*, and *Pseudomonas aeruginosa* are usually resistant via the PhoP-PhoQ and PmrA-PmrB pathways. Molecular evolutionary analysis of *mcr* genes was undertaken to show relative relatedness across the ten main lineages. Understanding the molecular determinants of resistance to polymyxins may help develop suitable and effective methods for detecting polymyxin resistance determinants and the development of novel antimicrobial molecules.

## 1. Introduction

The polymyxin antibiotics colistin and polymyxin B have been recently revitalized as bactericidal drugs due to the increase in bacterial resistance to many commonly used antibiotics [1,2]. Polymyxins were originally derived from the bacterium *Paenibacillus polymyxa* as the products of fermentation in the form of amphipathic lipopeptide molecules [3]. Polymyxins were discovered in the 1940s to be cyclic lipodecapeptide antibiotics [4] and recognized for therapeutic use in the 1950s [4,5]. Polymyxins contain conserved components that consist of a d-Phe^6^-l-Leu^7^ segment, an N-terminal fatty acyl chain separated by cationic residues (l-α-γ-diaminobutyric acid (Dab)), and segments of the polar amino acid threonine (Thr) [6,7]. Polymyxins target the negatively charged outer membrane lipopolysaccharides (LPSs) of Gram-negative bacteria [6].

There are five (A to E) types of polymyxins; however, only polymyxin B and polymyxin E (colistin) are available for clinical use. The difference between polymyxin B and E is a result of the replacement of the amino acid D-phenylalanine at the sixth position in polymyxin B with leucine in colistin [6,7]. Both polymyxins are heterogeneous assemblies of chemically related molecules that differ from each other at the N-terminus fatty acyl group [8,9]. For polymyxin B, the major fatty acyl groups are (S)-6-methyloctanoyl for polymyxin B1 and 6-methylheptanoyl for polymyxin B2, whilst in colistin, the main fatty acyl groups are (S)-6-methyloctanoyl for colistin A and 6-methylheptanoyl for colistin B [8,9]. However, the exact proportions of each of these components can vary among different manufacturers or different batches of the same manufacturer [8,9,10]. The use of polymyxins was limited partly due to their toxicity. However, polymyxins are now considered to be last resort antibiotics [11], even though bacteria are becoming resistant to polymyxin B and colistin [11,12].

The aim of this literature review was to decipher the evolving mechanisms of resistance to polymyxins among pathogens of clinical significance. A review of the literature was undertaken with the following keywords and Boolean search criteria “(polymyxins) AND (molecular mechanism of resistance) AND (mcr gene) AND (intrinsic resistance) AND (molecular evolution)”. Two search engines were used: Scopus and Medline. The articles were restricted to original research articles, reviews, or case reports published in English with full versions available online. After obtaining relevant articles, their references were investigated for any additional articles that were pertinent to the aim of this study.

## 2. Rate of Resistance to Polymyxins among Pathogens of Clinical Significance

A constant and ongoing threat to public health is the global emergence of bacteria with multidrug resistance (MDR) and pan-drug resistance (PDR), rendering most or all commercially available antibiotics ineffective [12,13]. Among the commonly resistant pathogens are Gram-negative bacteria [12]. The World Health Organization (WHO) has identified several bacteria of critical importance due to their increasing resistance to antibiotics, namely, MDR *Acinetobacter baumannii*, carbapenem-resistant *Enterobacteriaceae*, and MDR *Pseudomonas aeruginosa* [12]. To treat such pathogens, polymyxins have been considered as last-resort antibiotics [14]. However, resistance to polymyxins has been reported to be frequent among clinical isolates of many of these Gram-negative bacteria [15,16]. The emergence of mobilized colistin resistance (*mcr*)-containing plasmids and chromosomally integrated *mcr-1* that mediate colistin resistance have generated a significant sense of global public health alarm, leading to concerns about the future effectiveness of colistin [17,18,19,20,21]. 

The rate of polymyxin resistance varies depending on the bacterial species and geographic location, and the exact rate of polymyxin resistance can only be determined using specific molecular studies or surveillance systems [22,23,24,25]. Colistin resistance of *A. baumannii* was first reported in 1999, and its rate of resistance has gradually increased over the past few decades [26]. In 2013, the European Antimicrobial Resistance Surveillance Network (EARS-Net) collected data from 17 countries in Europe and found an average resistance rate of 5%. A retrospective study from French Guiana in 2017 found a similar resistance rate at 4.4% to colistin [27]. Another multi-center epidemiological surveillance study, MARATHON, reported a colistin resistance rate of 1.9% in *A. baumannii* isolates in Russia between 2013 and 2014 [28]. However, Italy and Greece reported high rates of colistin resistance *A. baumannii* isolates, with over 80% resistance [29]. A study conducted in South Korea of 265 isolates of *Acinetobacter* spp. collected from tertiary-care hospitals between 2002 and 2006 reported an overall resistance rate of 27.9% (74/265) and 18.1% (48/265) to colistin and polymyxin B, respectively [30]. Another study based on SENTRY data from Korea revealed a high colistin resistance rate of 30.6% in *A. baumannii* isolates between 2006 and 2009 [15]. Furthermore, in Brazil, the resistance rates of *A. baumannii* were 81.5% in 2016 and 78.5% in 2021. The minimum inhibitory concentrations (MICs) varied from 4 to 64 μg/mL for polymyxin B and from 16 to 128 μg/mL for colistin in 2016, while the colistin MIC was 128 μg/L in 2021 [31,32]. 

For *P. aeruginosa*, an increasing trend in resistance has been reported in the EARS-Net surveillance study, with a 1% to 4% rise in colistin resistance in Europe from 2013 to 2016 [33]. In 2016, most of the colistin-resistant isolates were from Italy and Greece. A rate of 1–*7*% colistin resistance has been described in China [34,35]. A resistance rate of 11.5% to polymyxins B has been described among *P. aeruginosa* strains isolated from keratitis in Australia and India [36]. 

For *Klebsiella pneumoniae*, an increasing rate of colistin resistance has been observed since the first report of colistin resistance in 2004 [37]. There was an increasing rate of colistin resistance in *K. pneumoniae* in Tunisia from 3.6% in 2002 to 9.7% in 2013 [22]. In Europe, the resistance rate of *K. pneumoniae* to polymyxins increased from 1.1% to 2.2% between 2003 and 2009 [23]. The European Antimicrobial Resistance Surveillance Network (EARS-Net) reported that in 2014, the highest resistance rate of 25.8% to polymyxins was found in Greece [38]. In this report, colistin resistance was at 29% among all carbapenem-resistant *K. pneumoniae* strains and 3% among carbapenem-susceptible isolates [38]. A study conducted in 2018 reported colistin resistance rates of 27% and 43% among carbapenem-resistant *K. pneumoniae* isolates in Dubai and Italy, respectively [39].

Data from an epidemiological survey of 25 provinces in China found colistin resistance at 1.4%, 2.9%, 2.4%, and 4% in *K. pneumoniae*, *Enterobacter cloacae*, Citrobacter freundii, and *E. coli*, respectively, among 1801 carbapenem-resistant *Enterobacteriaceae* (CRE) clinical strains between 2012 and 2016 [24].

According to a study on the global prevalence of colistin resistance, the highest rate of colistin resistance was observed in *K. pneumoniae* isolates in 2020, with a resistance rate of 12.9% (4 out of 31) [40]. In contrast, the colistin resistance rate among *K. pneumoniae* isolates studied from 2015 to 2019 was 2.9%. The study also found that *K. pneumoniae* isolates from Thailand had the highest rate of colistin resistance at 19.2%, while South Korea had the lowest prevalence of colistin resistance at 0.8% [40]. Reports from India indicated the rates of colistin resistance were 1.3% among a total of 21.3% ICU isolates and 38.4–46.6% among other clinical isolates of *K. pneumoniae* in 2020 and 2021 [41,42].

Contrary to *A. baumannii* and K. pneumoniae, colistin resistance is not so common in clinical isolates of *Escherichia coli.* For example, the colistin resistance rate was 0.2% and 0.9% among clinical and commercial meat specimens, respectively, between 2010 and 2014, and 1.1% to 8.7% in *E. coli* between 2012 and 2015, respectively, in Taiwan [43]. In China, resistance rates to polymyxin B and colistin of 0.3% to 7.3% among clinical isolates of *E.coli* have been reported in reports from 2013 to 2016 and 2021, respectively [44,45]. 

A study in 2022 demonstrated that 15% of carbapenem-resistant *Enterobacterales* isolates in India showed resistance to colistin [46]. The geographical distribution of significant polymyxin resistance strains is shown in a geospatial map in Figure 1.

## 3. Mechanisms of Resistance to Polymyxins in Different Bacteria

Polymyxin-resistant bacteria can cause serious infections and pose a significant threat to public health [7,47]. Bacterial cells have evolved various mechanisms to develop resistance toward polymyxins, including modification of the outer membrane, alterations in lipid A, and the use of efflux pumps. Cross-resistance between colistin and polymyxin B has been reported. Two primary mechanisms are responsible for the development of polymyxin resistance: (i) intrinsic mechanisms and (ii) acquired plasmid-mediated *mcr*-based mechanisms.

### Evolving Intrinsic Mechanisms of Resistance to Polymyxins

In the past, resistance to polymyxins was primarily attributed to mutations in chromosomal genes linked to the synthesis of lipopolysaccharides (LPSs) [48]. This resistance is commonly associated with two-component systems, often PhoPQ and PmrAB, as well as sets of regulatory genes such as the operon *arnBCADTEF* (also known as *pmrHFIJKLM*), *crrAB*, *mgrB*, and *pmrE* [49,50,51,52].

Cationic antibiotics such as polymyxins trigger the loss of cations (Ca^2+^ and Mg^2+^) from the negatively charged outer membrane of Gram-negative bacteria. This stress, as well as a high Fe^3+^ concentration and acidic pH, can activate the two-component systems PhoPQ and PmrAB and the *arnBCADTEF* cascade [14,53,54]. PhoPQ and PmrAB induce the synthesis of phosphoethanolamine (PEA) and/or 4-amino-4-deoxy-L-arabinose (L-Ara4N), which are then integrated into outer membrane LPS [53]. This addition of PEA or L-Ara4N provides additional cationic groups, leading to modifications in LPSs that neutralize the negative charge on the outer membrane, hindering further binding of colistin [53]. Increased expression of *arnBCADTEF* correlates with polymyxin resistance [55]. Mutations in *pmrB* cause high expression of *pmrC*, leading to the modification of lipid A with PEA in *K. pneumoniae* [56]. Also, exposure to chlorhexidine is associated with the development of colistin resistance in *K. pneumoniae* due to a point mutation in *pmrB* [57].

Furthermore, *mgrB* is involved in the feedback control of PhoPQ, and thus alterations in *mgrB* can also contribute to the development of polymyxin resistance. Mutations in *crrB* that induce *crrC* expression lead to hyper-expression of the *pmrAB* system and ultimately the development of resistance [58]. Additionally, the regulatory systems of colistin resistance involve *vprAB* in *Vibrio cholerae*, and *cprRS* and *parRS* in *P. aeruginosa*, which affect cation peptides in the outer membrane [59,60]. Common pathways of polymyxin resistance in *E. coli*, *K. pneumoniae*, and *P. aeruginosa* are shown in Figure 2.

#### Distinct Mechanisms of Intrinsic Polymyxin Resistance in *A. baumannii*

*A. baumannii* exhibits unique mechanisms of colistin resistance. The first mechanism involves the complete loss of lipooligosaccharide, which is caused by mutations in lipopolysaccharide peroxidation (LpX) genes (*lpxA*, *lpxC*, *lpxD*) and *vacJ* [61], responsible for lipid A synthesis. These mutations lead to permeability defects due to the encoding of acyltransferases, which are key enzymes in lipid A biosynthesis [62,63,64,65,66]. The second mechanism involves lipo-oligosaccharide modification by the addition of PEA or the transfer of L-Ara4N to the phosphate groups of lipid A [62,67]. There are two distinct pathways for the regulation of this modification. The first pathway involves the *pmrAB* operon, a two-component system that induces *pmrC*, which results in an LPS modification [2,61,64,65,68,69]. The second mechanism involves the insertion of the IS element ISAba125 into the transcriptional regulator H-NS family to increase the expression of *eptA* encoding for PEt that synthesizes PEA. This reduces the overall membrane electronegativity and so a reduction in membrane affinity for polymyxins [70,71]. Colistin resistance can also result from the loss of OmpW and production of DedA, as well as the expression of *eptA* in *Acinetobacter* spp. [72].

In 2022, a new two-component system (TCS) named StkSR was discovered in *A. baumannii* [73]. Deletion of *stkR* significantly increased the expression of *pmrA*, *pmrC*, and *pmrB*, leading to an increase in *pmrC* transcription and subsequent substitution of lipid A with PEA. There may be a regulatory relationship between the StkSR and the PmrAB systems based on the observed correlation in gene expression [73]. Distinct pathways of polymyxin resistance in *Acinetobacter* spp. are depicted in Figure 3. 

## 4. Plasmid-Mediated *mcr* Gene-Based Polymyxins

Mobilized colistin resistance, *mcr*, genes are mainly associated with bacterial plasmids. These play an important role in the spread of colistin resistance because of their transferability among different strains in different environments [28,44,53,74]. 

These *mcr* genes encode phosphoethanolamine-lipid A transferases [75,76] that mediate the addition of PEA to the lipid A of an LPS at the 1′ and 4′ positions, causing a significant reduction in the overall negative charge on the bacterial outer membrane [77,78]. This ultimately leads to the loss of binding affinity of an LPS to the cationic polymyxins and therefore resistance to their action [76,78]. 

These *mcr* genes have several variants. The nucleotide sequence of *mcr-1* recovered from different strains is highly conserved [79,80], while that of *mcr-2* is variable. Two variants *mcr-2.1* (MF176239) and *mcr-2.2* (MF176240) from different strains have nucleotide sequence similarities from 95.4% to 97.5% and 87.0% to 88.4%, respectively [80]. However, the other 33 identified variants of *mcr-2* (from 2.3 to 2.35) are very similar with limited mismatches of about 211-44 nucleotides [80]. 

The *mcr-3* variants are very similar to *mcr-1* with 45% nucleotide sequence similarity [81] and high protein similarity of 60% to each other [82]. To date, *mcr-3* has 42 variants, *mcr*-*3.1* to *mcr*-*3.42*, [79,82,83,84,85], which differ by only a few nucleotides. The *mcr* gene *mcr-4* has six variants, *mcr-4.1* to *mcr-4.6* [86,87,88,89,90], and *mcr-5* has four variants, *mcr-5.1* to *mcr-5.4* [91,92,93,94]. The Mcr-5 protein has very low amino acid sequence similarity of only 33–36% to Mcr-1, Mcr-2, Mcr-3, or Mcr-4 [88]. The *mcr-6.1* gene is 82.8% similar to *mcr-1* and *mcr-2* [95].

The proteins encoded by *mcr-7.1* have ∼70% identity to Mcr-3.1 and ∼30–45% similarity with other Mcr proteins [96], while Mcr-8 shows low similarity with Mcr-1 (31.08%), Mcr-2 (30.26%), Mcr-3 (39.96%), Mcr-4 (37.85%,), Mcr-5 (33.51%,), Mcr-6 (30.43%), and Mcr-7 (37.46%) [97]. *mcr-8.2* is a recently discovered variant of *mcr-8* [98]. *mcr-9* has been reported in the *Salmonella enterica* serotype Typhimurium [99], and the Mcr-9 protein is related to Mcr-3, Mcr-4, and Mcr-7, with the highest level of similarity with Mcr-3 (64.5% amino acid identity and 99.5% nucleotide similarity [99]). All Mcr proteins 1-9 have highly conserved catalytic and membrane-anchor domains, although these may not always be functionally interchangeable [74,100]. The *mcr-10* gene has a nucleotide sequence identity of 79.69% and an amino acid sequence identity of 82.93% to Mcr-9 [100]. Mcr-10 also shares significant amino acid identity with the product of the chromosomally encoded *mcr*-like PEt from *Buttiauxella* species [101]. 

To gauge the phylogenetic relatedness of *mcr* variants, *mcr* sequences were retrieved from the NCBI GenBank, and their sequences were aligned using ClustalW. A neighbor-joining phylogenetic tree was constructed using MEGA 11 [102] and visualized using iTOLv6 [103] (Figure 4).

### 4.1. Global Dissemination of mcr among Different Bacteria in Different Environments

It is believed that sporadic outbreaks of *mcr* occurred in Chinese food-producing livestock in 1980 [17]. Since that time, *mcr-1*-carrying bacterial strains have been reported in several countries among five of the seven continents across the globe [17,25,43,104,105,106] including China [25], India [107], Pakistan [108], Vietnam [109], Laos [110], USA [111], Italy [112], and Japan [79]. 

The transmission of *mcr* genes carrying pathogens could occur from animals to humans via direct contact with food animals and pets [113,114,115]. Also, reservoirs for *mcr-1*-carrying bacteria have been identified in public beaches [116], hospital sewage, wastewater treatment plants [117,118], rivers [115], and water wells in rural areas [119], as well as from houseflies and blowflies [120]. Although data from some studies suggests that flies might be intermediate vectors for transmission of *mcr-1*-containing bacteria between companion animals and humans [121], the exact route for the spread of *mcr-1* and the bacteria carrying *mcr-1* needs more thorough investigation.

Several species of *Enterobacteriaceae* possess *mcr-1*, such as *E. coli* where the gene is carried on IncI2 and IncX4 plasmids [122], *Enterobacter aerogenes* on an IncX4 plasmid [123], *E. cloacae* on an IncFI plasmid [123], *Cronobacter sakazakii* on an IncB/O plasmid [124], *Citrobacter freundii* on an IncHI2 plasmid [125], *C. braakii* on an IncI2-type plasmid, *K. pneumoniae* on an IncX4 plasmid [126], *Salmonella enterica* on IncHI2-like plasmids [127], *Shigella sonnei* on IncHI2-like plasmids [128], and *Raoultella ornithinolytican* on an IncHI2 plasmid [129]. Also, *mcr-1* variants have been identified in strains co-harboring *bla*_NDM-5_ that confers carbapenem resistance to *E. coli* [108]. The *mcr-1.1* gene has been found in the chromosome of *E. coli* and plasmid p16BU137 of *K. pneumoniae* from environmental isolates in China [76]. Further details of recently discovered *mcr* variants and their respective transposons and plasmids are given in Table 1.

In Australia, colistin resistance was reported among poultry isolates of *Aeromonas hydrophila*, *Alcaligenes faecalis*, *Myroides odoratus*, *Hafnia paralvei*, and *Pseudochrobactrum* spp. from a chicken processing unit in the state of Victoria [130]. Furthermore, *mcr-1* was found in association with incompatibility group IncI2 plasmids from isolates in the state of New South Wales (NSW) [131], and *mcr*-1.1 has been detected in *E. coli* [132]. Similarly, *mcr*-*1.1* and *mcr-3* were found among MDR isolates of *Salmonella enterica* 4 from human and animal sources in NSW [132,133]. An evolutionary analysis of multiple drug-resistant *Salmonella enterica* serovar 4 indicated that the spread of the *mcr-3* variant in lineages 1 and 3 was associated with overseas travel to Southeast Asia [84]. Lineage 1 included *mcr-3.1*- and *bla*_CTX-M-55_-positive isolates of *Salmonella enterica* sequence type 34 from Europe and Asia that were resistant to colistin and third-generation cephalosporins [81,84]. Whilst *mcr*-3.2 in lineage 3 was associated with IncHI2 pST3 and IncAC plasmids, wherein the colistin resistance genes were part of *dgkA* (diacylglycerol kinase) [84,134], which is a small transposable unit associated with IS elements circularized and integrated into *Enterobacterales* genomes [80].

### 4.2. Evolution of mcr Gene Variants from mcr-1 to mcr-10

In the current study, the phylogeny among *mcr* variants was determined using Molecular Evolutionary Genetics Analysis (MEGA 11) and is shown in Table 1. This shows the pair-end number of substitutions between *mcr-1* and *mcr-10*, with the number of base differences per site indicated. An estimate of evolutionary divergence between the sequences of *mcr-1* and *mcr-10.1* was performed using MEGA 11. Overall, the average divergence among *mcr* ranged from 52 ± 20% for *mcr-2* compared to all others to 69 ± 4% for *mcr-8*.

Moreover, phytogenic analysis of *mcr-3* also demonstrated that most occurred and evolved among *Aeromonas* species. This suggested the origin of *mcr*-*3* was *Aeromonas* species with gradual evolution and transmission of *mcr-3* variants to *E. coli* and *K. pneumoniae*, while other *mcr* gradually evolved among *E. coli* and *K. pneumoniae.* Interestingly, after the emergence of *mcr*-*4*, the identification of *mcr*-4.3 in *A. baumannii* represented a gradual evolution of *A. baumannii* against colistin with a distinct type of *mcr* gene in the form of a novel plasmid carrying *mcr*-*4.3* [135]. 

The analysis of evolutionary probabilities in *mcr* variants used a previously described method [136] using modified evolutionary probabilities (EPs) [137]. A user-specified tree topology was analyzed using the maximum likelihood method and the general time reversible model [138]. The evolutionary time depths used in the EP calculation can be obtained using the real-time [139] method. This analysis involved using the 10 nucleotide sequences of *mcr*. Codon positions included the first + second + third plus the noncoding positions. All positions containing gaps and missing data were eliminated (complete deletion option). The results, which represent the number of base differences per site for each *mcr* variant, are depicted in (Figure 5).

The probability of substitution of nucleotides to *mcr*-1 is demonstrated in Figure 5, which shows that the most likely substitution of adenine was with guanine (12%), of thymine was with cytosine (15%), of cytosine was with thymine (15%), and of guanine was with adenine (11%). The positions of substitution of nucleotides (A, T, G, and C from position 1 to 262 of different sites) for *mcr*-1 (*E. coli* strain ZZ1409 KU886144) are shown in Figure 6, respectively. In terms of positioning, cytosine (C) is predominately present at positions 1 to 257, followed by adenine (A) from positions 1 to 253, guanine (G) from positions 1 to 261, and thymine (T) from positions 5 to 261. In terms of probability and position of substitution, guanine was mostly likely to be present at position 27 with a probability of 0.95, and least likely to be present at position 28 with a probability of substitution of 0.007; thymine was most likely to be present at position 30 with a probability of 0.95 and least likely to be present at position 28 with a probability of 0.007; adenine was most likely to be present at position 220 with a probability of 0.94 and least likely to be present at position 27 with a probability of 0.007; cytosine was most likely to be present at position 160 with a probability of 0.93 and least likely to be present at position 262 with a probability of 0.014.

#### The Processes and Molecular Vehicles Responsible for the Transmission of *mcr* Variants

Studies have comprehensively analyzed the genetic environments of *mcr*-carrying genomes using bioinformatics tools such as Geneious R8 [140] and ISfinder software [141] to demonstrate the insertion of *mcr* variants. The structures of recently reported insertion sequences and the names of their associated transposons are given in Table 2.

Full genome sequencing and analysis for identification of replication origin (*oriC*) in *mcr-1*-harboring plasmids from colistin-resistant isolates have identified a novel hybrid IncI2/IncFIB plasmid pGD17-2 [142]. Moreover, the co-occurrence of pGD17-2 with plasmids pGD65-3, IncI2, and pGD65-5, IncX4 has been reported in a single drug-resistant isolate (GD65), and this co-occurrence might promote the dissemination of *mcr-1* under environmental selection pressure [142]. *mcr*-1 and other clinically significant resistant genes such as extended-spectrum β-lactamase (ESBL) *blaCTX-8* and *blaCTX-M-1* are related to globally identified sequence types ST10, ST46, and ST1638 in pathogenic strains of *E. coli* responsible for infections in humans and animals [143,144,145]. *E. coli* ST10 stains carrying *mcr-1* have been isolated from water at a public beach in the USA where the same ST10 strain had been isolated from an infected migratory Magellanic penguin with pododermatitis [143], suggesting that the ST10 strains carrying *mcr-1* can disseminate in the marine environment. *E. coli mcr-1*-positive environmental isolates have been isolated from German swine farms [146] and in diseased food animals in China [147], Italy, and France [148]. A plastidome analysis of *mcr*-*1* of *Enterobacterales* human isolates suggested that the spread of *mcr-1* among commensals such as *K. pneumoniae*, *E. coli*, and other clinical isolates could be facilitated by various promiscuous diverse plasmids [149]. 

Insertion sequences (ISs) or integrons can also facilitate the spread of *mcr*. An analysis of *mcr-1* from various sources using whole genome sequencing supported a single *mcr-1* mobilization event in IS*Apl1-mcr-1-orf*-IS*Apl1* transposon [150]. This transposon has been immobilized on different plasmids such as IncI2, IncHI2, and IncX4 [151]. Plasmids pGD65-3, IncI2, and pGD65-5, IncX4 contain two insertion sequences, IS*Ecp1* and IS*Apl1*, that facilitate the mobilization of *mcr-1* [142]. The insertion sequence IS*Apl1*, which originated in *Actinobacillus pleuropneumoniae*, is located upstream of *mcr-1* in the IncI2-type *mcr-1*-harboring plasmid Phnshp45 [74,152,153]. However, the IS*Apl1* element is not always found associated with *mcr-1* on most IncX4 plasmids [152,153,154]. A reason for this may be that the translocation of an *mcr-1-pap2* element by integration of an IS*Apl1* cassette (a member of the IS*30* family) [134,152] into plasmids such as pMCR1-IncI2, and pMCR1-IncX4 may induce the formation of circular intermediates by recognizing inverted repeat sequences, which ultimately results in loss of IS*Apl1* after integration of *mcr-1* [134,155,156].

The *mcr-2* gene is not associated with IS*Apl1*, but there are two IS*1595*-like insertion sequences predicted to surround *mcr-2* in the IncX4 plasmid pKP37-BE [157]. The short IS*1595*-like element carries a transposase gene flanked by two inverted repeats surrounding *mcr*-2. This transposase-encoding gene is similar (75% identity) to a fragment found in *Moraxella bovoculi* strain 58069, which suggests the origin of *mcr*-2 was from *M. bovoculi* [155]. The occurrence of duplicate target sites adjacent to a spacer sequence suggests that the spacer sequence is the most probable hot site in IncX4 plasmids for integration and transposition of *mcr-2* variants [158]. Transfer of *mcr-2* can occur through IS1595-containing transposons [155,156,158,159].

**Table 2 antibiotics-12-01597-t002:** Recently reported insertion sequences and transposon elements associated with *mcr* genes transmission.

*mcr* Variants	Insertion Sequences Structure	Transposon	Plasmids	Organism	Host (Isolated from)	Year of Discovery	References
*mcr* *-1*	(IS*Apl1-mcr-1-pap2-*IS*Apl1* and Tn*7511*)	Novel transposon Tn*7511*	IncI1 plasmid, pMCR-E2899	*E. coli* DH5α	Turkey meat	2022	[160]
*mcr* *-1*	Combination of IS*Apl1* and IS*91* (IS*Apl1*-*mcr*-*1*-IS*91*)	Chromosomal Tn*6330* transposon	IncI2 plasmid	*E. coli*	Community and hospital settings	2022	[74]
*mcr* *-1*	IS*26-mcr-1-PAP2*, and IS*APl1-mcr-1-PAP2* and IS*Ecp1-blaCTX₋M₋₅₅-mcr-1-PAP2*	---	IncI2, IncX4, and IncHI2 plasmids	*E. coli* and *Salmonella* spp.	Food products, food supply chain, and clinical samples	2021	[161,162]
*mcr* *-1.1*	IS*26-parA-mcr-1.1-pap2*	---	IncX4-type plasmid	*E. coli*	Dog feces	2020	[150]
*mcr* *-1*	*I* IS*Apl1-mcr-1-orf* IS*Apl1*	IS*Apl1* transposon	IncHI2 and IncX4 plasmids	*Enterobacteriaceae*	Livestock	2018	[163]
*mcr-1*	IS*Apl1-mcr-1-pap2-*IS*Apl1*	Tn*6330*	IncI2 and IncX4 plasmids	Novel *Moraxella* spp.	Pig	2018	[140]
*mcr* *-1*	*mcr-1-orf,* IS*Apl1-mcr-1-orf* and Tn*6330*	Novel transposon Tn*6330*	IncX4 and IncI2 plasmids	*E. coli*	Pig farms in China	2017	[162]
*mcr* *-2*	(IS*Ec69-mcr-2-ORF-*IS*Ec69*	Tn*7052*	IncX4 conjugative plasmid	*Moraxella osloensis*	---	2021	[164]
*mcr* *-2*	IS*Ec69-mcr-2-*IS*Ec69*	---	IncX4 plasmid	*M. bovoculi*	Pigs, pork and chicken meat, and humans	2017	[165]
*mcr* *- 3.1*	Tn*As2-mcr-3.1-dgkA-*IS*Kpn40*	Novel transposon Tn*6330*	pCP61-IncFIB plasmid	*E. coli*	Pigs	2021	[166]
*mcr* *-3.5*	IS*4321R-*Tn*As2-mcr-3.5-dgkA-*IS*15*	Novel transposon Tn*6330*	IncFIItype plasmid pCP55-IncFII	*E. coli*	Pigs	2021	[166]
*mcr* *-3.7*	Tn*As2-mcr-3.7-dgkA*-IS*26*	---	IncP1 plasmid	*E. coli*	Dogs	2020	[150]
*mcr* *-8*	IS*903B-ampC-hp-hphp-Giy-T-dgkA-baeS-copR-*IS*3-mcr-8-Gly-T-*IS*5*	_ Δ IS*66* transposases	IncFIA plasmid	*K. pneumoniae*	Patients from intensive care	2022	[167]
*mcr-8*	IS*903B*-*ymoA*-*inhA*-*mcr-8*-*copR*-*baeS*-*dgkA*-*ampC*	Composite transposon	pABC264-OXA-181 plasmid	*K. pneumoniae*	Patient with bacteremia	2022	[168]
*mcr-8.2*	IS*Ecl1*-*mcr-8.2*-*orf*-IS*Kpn26*	---	IncFII/FIA	*K. pneumoniae*	Patient’s Intestinal sample	2022	[169]
*mcr-9.1*	IS*903B*-*mcr-9.1*-*wbuC*-IS*26*	Tn*6360*	IncHI2/2A plasmid	*E. cloacae* complex	Clinical isolates	2022	[170]
*mcr* *-10*	IS*Kpn26* is present at upstream of *xerC-mcr-10* and an IS*26*	Transposon Tn*1722*	IncFIA plasmid	*Enterobacter roggenkampii*	Clinical isolate	2020	[101]
*mcr-10.1*	*hsdSMR-ISEc36-mcr-10.1-xerC*	---	IncFII_K_ plasmids	*K. pneumoniae*	Clinical isolates	2022	[170]

### 4.3. Methods for Detecting Polymyxin Resistance

As resistance to polymyxins is being reported frequently among different bacterial isolates from humans, animals, and the environment, affordable, accessible, and efficient diagnostic approaches are needed. The phenotypic determination of colistin-resistant isolates can be made by growing on media such as CHROMagar COL- APSE [171], SuperPolymyxin™ [172], and LBJMR [173], as well as using commercial automated MIC-determining instruments such as BD Phoenix, MicroScan, Vitek 2 [174], MICRONAUT- S [175], and Sensititre [176]. The rapid polymyxin NP test and its modifications [177], colispot [178] colistin MAC test [179], MIC Test Strip, MICRONAUT-MIC Strip [180], the UMIC System [181], and Sensitest Colistin [176] can also be used [174]. Eazyplex SuperBug kit [182] and Taqman/SYBR Green real-time PCR assays have been used for molecular identification of *mcr* genes that have yielded 100% specificity and sensitivity with a rapid turnaround time (<3 h) [183]. More advanced molecular techniques such as multi-loop-mediated isothermal amplification (multi-LAMP) assays can also be used for rapid detection of *mcr* genes [184]. Based on cost, sensitivity and specificity, turnaround time, and the skills required to perform the test, the use of culture media or the Rapid Polymyxin Nordmann–Poirel (RPNP) test are recommended for low-resourced laboratories, while Multiplex PCR or Taqman/SYBR Green real-time PCR assays along with RPNP or novel culture media are applicable for well-resourced laboratories [185,186]. 

To study the evolution in *mcr*-positive bacterial strains, different sequencing techniques can be used including Sanger sequencing and the identification of single nucleotide polymorphisms [187] for mutational analysis or identification of new *mcr*- variant(s) [188]. For detailed studies of intrinsic determinants of resistance, whole genome sequencing (WGS) [189], nanopore sequencing, and transposon-directed insertion site sequencing [165] can give insights into the interactions of genetic elements associated with polymyxins resistance. To study coevolution among pairs of *mcr* or multiple *mcr* elements within a single bacterial cell, *mcr*-coevolution assays could be used [165].

## 5. Conclusions

Polymyxins are a class of cationic polypeptide antibiotics that are involved in the disruption of LPS in Gram-negative pathogens. Polymyxins have been extensively used to treat infections after their initial approval for clinical use, but their use has been limited due to their nephrotoxicity and neurotoxicity. However, the development of bacterial strains resistant to many other types of antibiotics has led polymyxins to be reconsidered as a last-resort therapeutic option to treat MDR pathogens. This reuse has led to the re-emergence of resistance to polymyxins. Bacterial cells evolve resistance to polymyxins by modifying their LPSs, using antibiotic efflux pumps, or reducing the amount of LPSs produced. Mutations to existing genes or acquisition of mobile genetic elements such as transposons and insertion sequences play a major role in the development of resistance to polymyxins. *mcr* genes can be major role players in the global spread of colistin resistance because of their high mobility via plasmids. To date, several *mcr* variants (*mcr*-1 to *mcr-10*) have been reported among colistin-resistant pathogens. A detailed understanding of the molecular determinants underlying resistance to polymyxins, and ways of testing for resistance, can help to develop suitable and effective methods for detecting resistance to polymyxins, as well as aiding in the development of novel antimicrobials in the future. Therefore, ongoing research into the molecular determinants of polymyxin resistance is important for the development of effective strategies to combat antibiotic resistance and ensure the continued efficacy of polymyxins and other antimicrobial agents in clinical practice. 

## Figures and Tables

**Figure 1 antibiotics-12-01597-f001:**
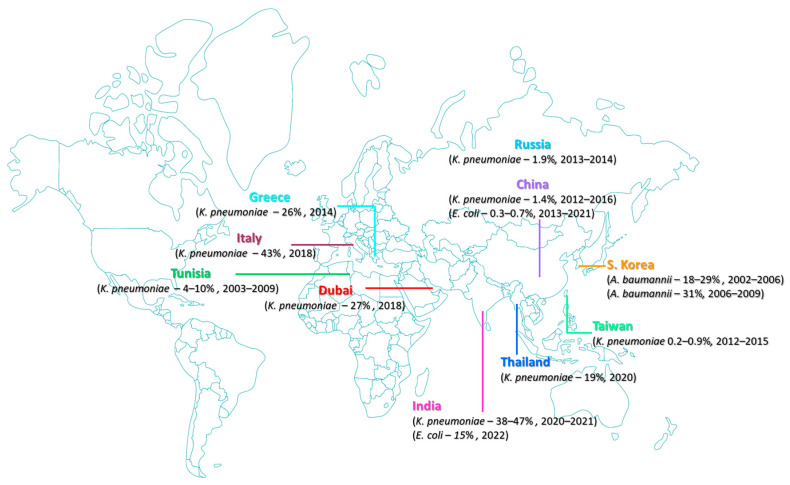
The global burden of significant polymyxin resistance from 2002 to 2022.

**Figure 2 antibiotics-12-01597-f002:**
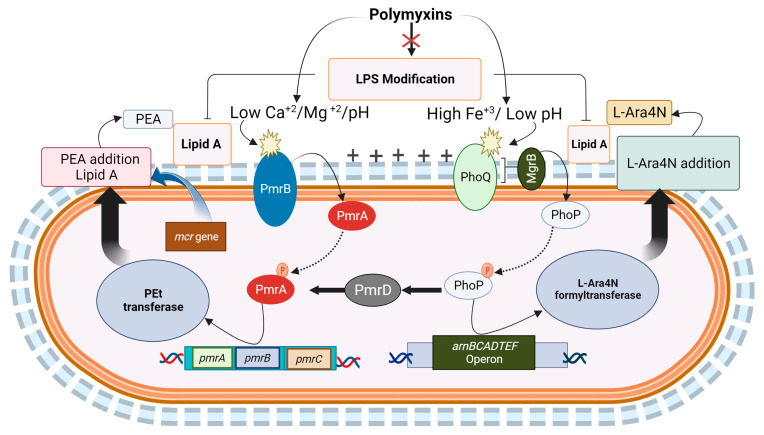
The mechanisms involved in the modification of lipopolysaccharides (LPSs) that contribute to polymyxin resistance in Gram-negative bacteria. In various bacteria such as *Salmonella* spp., *Escherichia coli*, *Klebsiella pneumoniae*, and *Pseudomonas aeruginosa*, the detection of different stress conditions triggers a response mediated by the histidine kinases PhoQ and PmrB. (The process of gene activation through phosphorylation is represented with dashed arrows, and the outcomes are represented with thick arrows). These stress conditions include the presence of cationic compounds like polymyxins, low concentrations of Mg^2+^ and Ca^2+^, acidic pH, and high concentrations of Fe3^+^. Activation of the two-component systems PhoP-PhoQ and PmrA-PmrB occurs because of sensing the flow of molecules and is represented with small black arrows. The subsequent activation of the *arnBCADTEF* and *pmrCAB* operons leads to the synthesis and addition of 4-amino-4-deoxy-L-arabinose (L-Ara4N) and phosphoethanolamine (PEA) to lipid A, respectively. The addition of PEA by phosphoethanolamine transferase (PEt) and L-Ara4N by L-Ara4N formyltransferase is denoted with thick black arrows, while PEA addition through *mcr* genes is represented with a thick blue arrow. Additionally, PhoP-PhoQ activation induces PmrAB through the product of the *pmrD* gene, which, in turn, activates *pmrA* to further trigger the *arnBCADTEF* operon. The PmrB and PhoQ activation is denoted with star-like symbols. Polymyxin resistance is also associated with the inactivation of MgrB, a negative regulator of the PhoP-PhoQ system. Amino acid substitutions in MgrB result in its inactivation, leading to overexpression of the *phoP*-*phoQ* operon and subsequent activation of the *pmrHFIJKLM* operon, which ultimately leads to the production of L-Ara4N.

**Figure 3 antibiotics-12-01597-f003:**
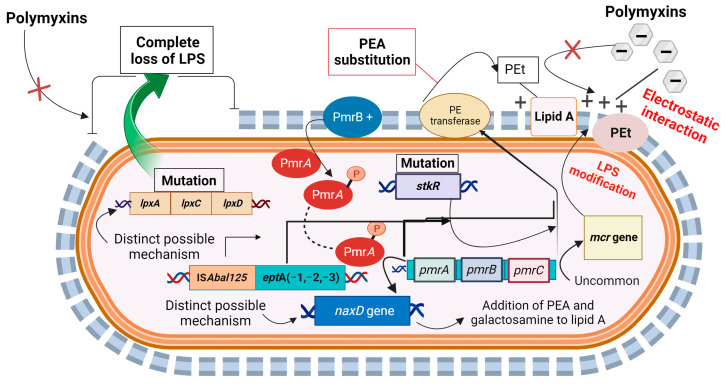
Distinct mechanisms of polymyxin resistance in *Acinetobacter baumannii* attributed to alterations in either PmrB or the response regulator PmrA, both of which are involved in activating the PmrAB two-component system. This activation leads to the upregulation of the *pmrCAB* operon and the *naxD* gene (as distinct pathways). The upregulation of the *pmrCAB* operon promotes the addition of PEA to lipid A, while the translation of *naxD* produces NaxD deacetylase, which is required for galactosamine addition to Lipid A. Another distinct possible pathway leading to PEt overproduction in *A. baumannii* involves IS*AbaI* insertion element integration upstream of an *eptA* isoform. Additionally, a mutation in *lpxACD* causes total loss of LPS *Acinetobacter* spp., which is a distinctive pathway of polymyxin resistance. Furthermore, a recently discovered novel pathway of polymyxin resistance in *A. baumannii* involves the deletion of *stkR*, which significantly increases the expression of *pmrA*, *pmrC*, and *pmrB* and ultimately increases *pmrC* transcription and the subsequent substitution of lipid A with PEA. Additionally, an uncommon pathway of Lipid A modification involves the mobile colistin resistance gene *mcr*, which encodes PEt, and the subsequent addition of PEA to LPSs.

**Figure 4 antibiotics-12-01597-f004:**
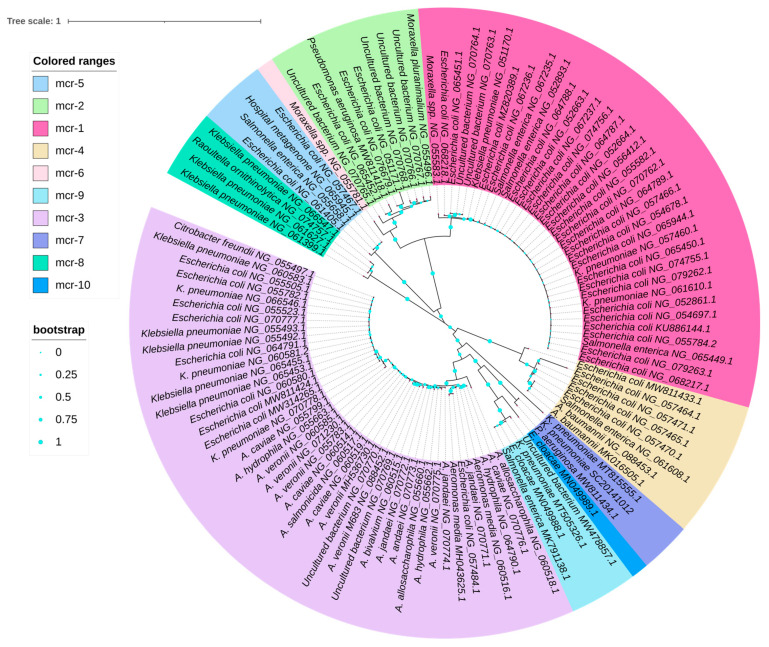
The phylogenetic relationship among the *mcr*-1 to *mcr*-10 variants using the neighbor-joining phylogenetic tree using the Kimura parameter with 1000 bootstraps using MEGA10, and visualised using iTOLv5 (Interactive Tree Of Life).

**Figure 5 antibiotics-12-01597-f005:**
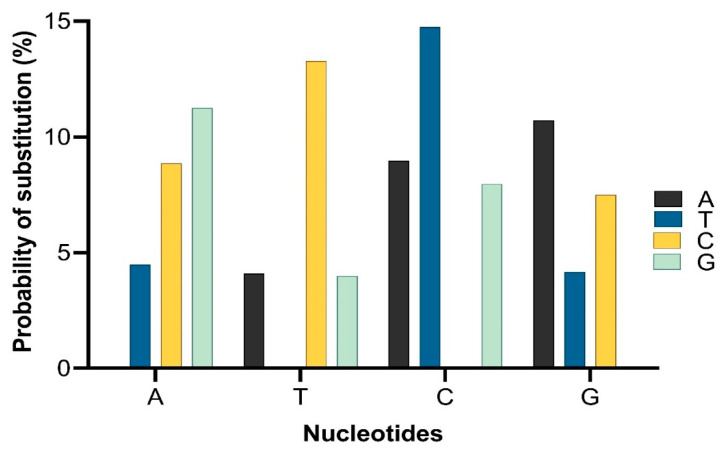
The probability of substitution of one base for another base. Substitution patterns and rates were estimated using the general time reversible model [1]. The maximum log-likelihood for this computation was 2655.269. This analysis involved all 10 nucleotide sequences of *mcr*. Codon positions included were 1st + 2nd + 3rd + noncoding. All positions containing gaps and missing data were eliminated (complete deletion option).

**Figure 6 antibiotics-12-01597-f006:**
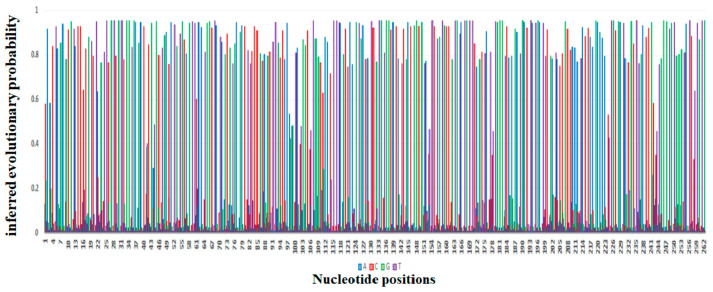
Depiction of the evolutionary probabilities of nucleotide substitution with respect to positions 1 to 262 for *mcr-1* in *Escherichia coli* strain ZZ1409 KU886144.

**Table 1 antibiotics-12-01597-t001:** The evolutionary divergence among *mcr* variants (*mcr-1* to *mcr-10*) (a score of 1 indicates no divergence between variants; a score of 0 indicates complete divergence).

	*mcr* Gene Number									
*mcr* gene number and source	1	2	3	4	5	6	7	8	9	10
*mcr-1 Escherichia coli* KU886144.1		0.18	0.67	0.57	0.54	0.22	0.47	0.68	0.71	0.71
*mcr-2 Pseudomonas aeruginosa* MW811418.1	0.18		0.68	0.58	0.56	0.12	0.49	0.69	0.7	0.72
*mcr-3 Escherichia coli* MW811424.1	0.67	0.68		0.62	0.75	0.68	0.7	0.76	0.38	0.38
*mcr-4 Escherichia coli* MW811433.1	0.57	0.58	0.62		0.56	0.58	0.49	0.65	0.65	0.61
*mcr-5.1 Salmonella enterica* NG055658.1	0.54	0.56	0.75	0.56		0.55	0.43	0.64	0.72	0.73
*mcr-6.1 Moraxella* sp. NG055781.1	0.22	0.12	0.68	0.58	0.55		0.51	0.72	0.72	0.74
*mcr-7 Pseudomonas aeruginosa* MW811434.1	0.47	0.49	0.7	0.49	0.43	0.51		0.65	0.71	0.68
*mcr-8 Klebsiella pneumoniae* MT815555.1	0.68	0.69	0.76	0.65	0.64	0.72	0.65		0.69	0.72
*mcr-9* Uncultured bacterium MW478857.1	0.71	0.7	0.38	0.65	0.72	0.72	0.71	0.69		0.22
*mcr-10.1 Enterobacter cloacae* MN044989.1	0.71	0.72	0.38	0.61	0.73	0.74	0.68	0.72	0.22	
**Average evolutionary divergence**	**0.53**	**0.52**	**0.62**	**0.59**	**0.61**	**0.54**	**0.57**	**0.69**	**0.61**	**0.61**
**Standard Deviation**	**0.20**	**0.23**	**0.14**	**0.05**	**0.11**	**0.23**	**0.11**	**0.04**	**0.18**	**0.19**

## Data Availability

Not applicable.

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
