# Peer review of "A Review of Resistance to Polymyxins and Evolving Mobile Colistin Resistance Gene (*mcr*) among Pathogens of Clinical Significance"

_antibiotics, 2023, doi:10.3390/antibiotics12111597_

Round 1

Reviewer 1 Report

Comments and Suggestions for Authors

This review summary the intrinsic resistance determinants and the mobile colistin resistance gene (mcr) among pathogens of clinical significance. It should be checked carefully for typos and sentences. Some as follows:

1.     Line 78-80,” Moreover, in Brazil A. baumannii resistance rates of 81.5% and 78 37.5%, with MICs ranging from 4 to 64 μg/ml and 16 to 128 μg/ml, polymyxin B and 128 μg/L against colistin in 2016 and 2021, respectively [31,32].” It is confusing. Please rephrase it and check others throughout the manuscript.

2.     Line 121-122, “Figure 1. Represents the global burden of significant polymyxins resistant strains reported since 2012 to 2021.” A study of 2022 in the figure.

3.     Line 134-135, “including PhoPQ and PmrAB, as well as sets of regulatory genes such as the operons pmrHFIJKLM, crrAB, mgrB and pmrE”.  When a gene name used, it should be italic, while protein names should be capitalized the first letter. Please check throughout the manuscript.

4.     Line 160-160, “In various species such as Salmonella spp., Escherichia coli, Klebsiella pneumoniae and Pseudomonas aeruginosa, and Salmonella spp.” Two “Salmonella spp”. Many other typos in the manuscript.

5.     Line 241-242, “with the highest level of similarity with MCR-3 (64.5% amino acid identity and 99.5% nucleotide similarity)”. Please check the number.

6.     Line 314-317, duplicated.

Author Response

Comment 1: Line 78-80,” Moreover, in Brazil A. baumannii resistance rates of 81.5% and 78 37.5%, with MICs ranging from 4 to 64 μg/ml and 16 to 128 μg/ml, polymyxin B and 128 μg/L against colistin in 2016 and 2021, respectively [31,32].” It is confusing. Please rephrase it and check others throughout the manuscript.

Response: Rephrased as “Furthermore, in Brazil, the resistance rates of A. baumannii were 81.5% in 2016 and 78.5% in 2021. The minimum inhibitory concentrations (MICs) varied from 4 to 64 μg/ml for polymyxin B and from 16 to 128 μg/ml for colistin in 2016, while the colistin MIC was 128 μg/L in 2021 [31,32]”

We have also checked for confusing sentence structure elsewhere in the manuscript and made appropriate changes

Comment 2: Line 121-122, “Figure 1. Represents the global burden of significant polymyxins resistant strains reported since 2012 to 2021.” A study of 2022 in the figure.

Response: corrected to “Figure 1. The global burden of significant polymyxin resistance from 2002 to 2022.” – to account for the reference that includes data from 2002-2006, and the 2022 reference.

Comment 3: Line 134-135, “including PhoPQ and PmrAB, as well as sets of regulatory genes such as the operons pmrHFIJKLM, crrAB, mgrB and pmrE”.  When a gene name used, it should be italic, while protein names should be capitalized the first letter. Please check throughout the manuscript.

Response: The names of genes and operons are italicised, and proteins are written with first letter capitalised throughout the manuscript.

Comment 4:   Line 160-160, “In various species such as Salmonella spp., Escherichia coliKlebsiella pneumoniae and Pseudomonas aeruginosa, and Salmonella spp.” Two “Salmonella spp”. Many other typos in the manuscript.

Response: Thank you there is one Salmonella spp. It is corrected as “In various species such as Salmonella spp., Escherichia coliKlebsiella pneumoniae and Pseudomonas aeruginosa.”

We have also carefully reviewed the manuscript and made changes for typos throughout.

Comment 5:     Line 241-242, “with the highest level of similarity with MCR-3 (64.5% amino acid identity and 99.5% nucleotide similarity)”. Please check the number.

Response: checked and confirmed.

Comment 6: Line 314-317, duplicated.

Response: deleted

Reviewer 2 Report

Comments and Suggestions for Authors

The authors provide a comprehensive review of polymyxin resistance in bacteria, covering its prevalence, molecular mechanisms, and molecular evolutionary analysis, among other topics. While the authors conducted a good job in thoroughly reviewing prior research, much of the content has already been summarized in previous review articles, making it difficult to discern the unique value of this paper. The molecular evolutionary analysis of the mcr genes is the most novel aspect of this review. However, there is a lack of detailed description of the analysis methods and the explanation of the results, which is the most critical area that needs improvement.

Below, I will detail the areas that necessitate improvement. I encourage the authors to address these concerns to enhance the rigor and academic significance of the paper.

【Section 3】

overall: The notation for pEtN, PEtN, and PetN should be standardized.

Line 172: If "prmHFIJKLM" is a synonym for "arnBCADTEF", it should be standardized to avoid any confusion.

line 191: According to the reference, it seems that deletion, rather than mutation, is considered to be correct.

【Fig.2】

1. Please confirm if the representation method of arrows, such as thickness or color gradient, has any significance. Additionally, it would assist in understanding the figure better if the meanings of the arrows were documented in the legend or figure.

2. Please confirm the accurate localization depending on whether the protein is a transmembrane or globular.

3. The meanings of star-like symbols attached to pmrB and PhoQ should be explained.

4. Depending on the protein and gene, the notation (capitalization, italics, etc.) should be corrected.

5. The diagram should illustrate the relationship between MgrB and PhoQ.

6. "PetN addition Lipid A" on the right side of the figure might be an error for "L-Ara4N".

7. The "LPS Modification" would be the attachment of pEtN or L-Ara4N to Lipid A. It might be confusing that they are shown separately in the figure.

8. There needs to be a clear differentiation between the representation of proteins (circles) and genes (rectangles).

【Fig.3】

1. It would be better to explain naxD and galactosamine in the text.

2. The meaning of the arrow connecting the "StkR gene" to "Novel mechanism" is unclear. Also, the description "Novel mechanism" seems inappropriate in this figure.

3. According to the reference, "ISAbal" might be an error for "ISAba125".

【Section 5】

Line 296-298: The authors states, "This shows the pairwise number of substitutions between mcr-1 to mcr-10, with the number of base differences per site indicated," but figure does not apoear to show the pairwise number of substitutions.

Line 310: It is necessary to specify the context of "10 nucleotide sequences."

Line 320-322: The authors states, "mcr-2 exhibits a significant divergence with a base pair difference of approximately 0.77 out of 10. Furthermore, mcr-3 and mcr-4 consistently show a divergence of about 0.68 and approximately 5.8 base differences per site, respectively." However, it is unclear where in the graph this conclusion is being drawn from.

Line 323: The meaning of the phrase "gradual increase" is unclear.

Line 328: The authors states, "Phylogeny of mcr genes depicts the greater evolution and transmission of mcr-3 followed by mcr-1". It is a fact that there should be many transmission as the mar-3 and 1 gene has been detected in numerous isolates. Nonetheless, the evidence to support the inference of "greater evolution" has not been demonstrated here.

【Fig.5】

1. It would be better to adjust the decimal places for visibility.

2. I think the words "Axis Title" should be deleted.

【Fig.6】

1. The expressions "row" and "column" do not seem appropriate in a bar chart.

2. I don't think the results have been interpreted at all. An explanation should be added in the text as to what this graph shows.

【Fig.7】

1. I don't think the results have been interpreted at all. An explanation should be added in the text as to what this graph shows.

Additionally, there are numerous formatting errors such as incorrect italicization and inconsistent capitalization of abbreviations. Please make the necessary corrections.

Author Response

Comment 1: 【Section 3】overall: The notation for pEtN, PEtN, and PetN should be standardized.

Response: phosphoethanolamine has been standardized to PEA. Phosphoethanolamine transferase is now standardized as PEt.

Comment 2: Line 172: If "prmHFIJKLM" is a synonym for "arnBCADTEF", it should be standardized to avoid any confusion.

Response: pmrHFIJKLM is italicised as suggested.

Comment 3: line 191: According to the reference, it seems that deletion, rather than mutation, is considered to be correct.

Response: Changed throughout as suggested.

Comment 4: [Fig.2】

  1. Please confirm if the representation method of arrows, such as thickness or color gradient, has any significance. Additionally, it would assist in understanding the figure better if the meanings of the arrows were documented in the legend or figure.

Response:  we have changed the figure legend to “Figure 2. The mechanisms involved in the modification of lipopolysaccharide (LPS) that contribute to polymyxin resistance in Gram-negative bacteria. In various bacteria such as Salmonella spp., Escherichia coli, Klebsiella pneumoniae and Pseudomonas aeruginosa, the detection of different stress conditions triggers a response mediated by the histidine kinases PhoQ and PmrB. (The process of gene activation through phosphorylation is represented with dashed arrows and the outcomes are represented with thick arrows). These stress conditions include the presence of cationic compounds like polymyxins, low concentrations of Mg2+ and Ca2+, acidic pH, and high concentrations of Fe3+. Activation of the two-component systems (TCSs) PhoP-PhoQ and PmrA-PmrB occurs because of sensing the flow of molecules and is represented with small black arrows. The subsequent activation of the arnBCADTEF and pmrCAB operons leads to the synthesis and addition of 4-amino-4-deoxy-L-arabinose (L-Ara4N) and phosphoethanolamine (PEA) to lipid A, respectively. The addition of PEA by phosphoethanolamine transferase (PEt) and L-Ara4N by L-Ara4N formyltransferase is denoted by thick black arrows, while PEA addition through mcr genes is represented with a thick blue arrow. Additionally, PhoP-PhoQ activation induces PmrAB through the product of the pmrD gene, which, in turn, activates pmrA to further trigger the arnBCADTEF operon. The PmrB and PhoQ activation is denoted by star-like symbols. Polymyxin resistance is also associated with the inactivation of MgrB, a negative regulator of the PhoP-PhoQ system. Amino acid substitutions in MgrB result in its inactivation, leading to overexpression of the phoP-phoQ operon and subsequent activation of the pmrHFIJKLM operon, which ultimately leads to the production of L-Ara4N.” to include information on the meaning of the arrows.

  1. Please confirm the accurate localization depending on whether the protein is a transmembrane or globular.

Response: Confirmed by reviewing Uniprot and references, and altered where required.

  1. The meanings of star-like symbols attached to pmrB and PhoQ should be explained.

Response: Star-like symbols represent activation of PmrB and PhoQ. This is now described in figure legend as:

“The PmrB and PhoQ activation is denoted by star-like symbols.”

  1. Depending on the protein and gene, the notation (capitalization, italics, etc.) should be corrected

Response: This has corrected. All protein now have first letter are capitalized and all genes are italicized

  1. The diagram should illustrate the relationship between MgrB and PhoQ.

Response: Relationship between MgrB and PhoQ now denoted with relationship sign using

  1. "PetN addition Lipid A" on the right side of the figure might be an error for "L-Ara4N".

Response: Corrected

  1. The "LPS Modification" would be the attachment of pEtN or L-Ara4N to Lipid A. It might be confusing that they are shown separately in the figure.

Response: As PEA (previously pEtN) and L-Ara4N are separate proteins and activated differently we have chosen to separate their actions as we found it confusing to try to show these combined.

  1. There needs to be a clear differentiation between the representation of proteins (circles) and genes (rectangles).

Response: Now all proteins are represented as circles and genes as rectangles

Comment 5:【Fig.3】

  1. It would be better to explain naxD and galactosamine in the text.

Response: Explained in text as: while translation of naxD produces NaxD deacetylase which is required for galactosamine addition to Lipid A. – line 206-207

  1. The meaning of the arrow connecting the "StkR gene" to "Novel mechanism" is unclear. Also, the description "Novel mechanism" seems inappropriate in this figure.

          Response: notation for StkR gene mutation as novel mechanism is removed in figure 3.

  1. According to the reference, "ISAbal" might be an error for "ISAba125".

          Response: Apologies – corrected to "ISAba125" in figure 3.

Comment 6:【Section 5】

Line 296-298: The authors states, "This shows the pairwise number of substitutions between mcr-1 to mcr-10, with the number of base differences per site indicated," but figure does not appear to show the pairwise number of substitutions.

Response: Apologies - this has been changed. We have now removed the previous Figure 5 to which this should have referred, and substituted new Table 1. We believe this gives the information in a more user friendly format.

Comment 7: Line 310: It is necessary to specify the context of "10 nucleotide sequences."

Response: changed to “This analysis involved using the 10 nucleotide sequences of mcr”. Line 315

Comment 8: Line 320-322: The authors states, "mcr-2 exhibits a significant divergence with a base pair difference of approximately 0.77 out of 10. Furthermore, mcr-3 and mcr-4 consistently show a divergence of about 0.68 and approximately 5.8 base differences per site, respectively." However, it is unclear where in the graph this conclusion is being drawn from.

Response: This has been removed from the manuscript.

Comment 9: Line 323: The meaning of the phrase "gradual increase" is unclear.

Response: This section ha been removed from the manuscript. 

Comment 10: Line 328: The authors states, "Phylogeny of mcr genes depicts the greater evolution and transmission of mcr-3 followed by mcr-1". It is a fact that there should be many transmission as the mcr-3 and 1 gene has been detected in numerous isolates. Nonetheless, the evidence to support the inference of "greater evolution" has not been demonstrated here.

Response: This section has been removed from the manuscript. 

Comment 11:【Fig.5】

  1. It would be better to adjust the decimal places for visibility.

Response: The original Figure 5 has been deleted.

  1. I think the words "Axis Title" should be deleted.

Response: The original figure 5 has been deleted.

Comment 12:【Fig.6】

  1. The expressions "row" and "column" do not seem appropriate in a bar chart.

Response: word row and column has been removed – now Figure 5.

  1. I don't think the results have been interpreted at all. An explanation should be added in the text as to what this graph shows.

Response: Results are now interpreted in text as following:

The probability of substitution of nucleotides to mcr-1 is demonstrated in Figure 5 which shows that the most likely substitution of adenine was with guanine (12%), of thymine was with cytosine (15%), of cytosine was with thymine (15%), and of guanine was with adenine (11%).

Comment 13:【Fig.7】

  1. I don't think the results have been interpreted at all. An explanation should be added in the text as to what this graph shows.

Response: Figure 7 in is now interpreted in text as following:

In terms of positioning, cytosine (C) is predominately present at positions 1 to 257, followed by adenine (A) from position 1 to 253, guanine (G) from position 1 to 261 and thymine (T) 5 to 261. In terms of probability and position of substitution, guanine was mostly likely to be present at position 27, with a probability of 0.95 and least likely to be present at position 28 with a probability of substitution of 0.007; thymine was most likely to be present at position 30 with a probability of 0.95 and least likely to be present as position 28 with a probability of 0.007; adenine was most likely to be present at position 220 with a probability of 0.94 and least likely to be present at maximum probability of 0.93; cytosine was most likely to be present at position 160 with a probability of 0.93 and least likely to be present at position 262 with a probability of 0.014.

 Comment 15: Additionally, there are numerous formatting errors such as incorrect italicization and inconsistent capitalization of abbreviations. Please make the necessary corrections.

Response: review and corrected.

Round 2

Reviewer 1 Report

Comments and Suggestions for Authors

References: gene name, bacterial species should be italic.

Reviewer 2 Report

Comments and Suggestions for Authors

Thank you for responding to the large volume of comments.